# A comparison of long-term maternal mortality associated with pathologic placental separation: Highlighting possible trends and mechanisms

Sona Jasani[1], Atalay Demiray[2]*, Julia Stevenson[3], Conrad Krawiec[4]

1 Department of Obstetrics, Gynecology and Reproductive Sciences, Division of Obstetric Specialties and Midwifery, Yale School of Medicine, New Haven, Connecticut, United States of America, 2 Department of Health Policy and Management, Yale School of Public Health, New Haven, Connecticut, United States of America, 3 Yale School of Medicine, New Haven, Connecticut, United States of America, 4 Department of Pediatrics, Division of Critical Care Medicine, Pennsylvania State Health, Hershey, Pennsylvania, United States of America

◉ These authors contributed equally to this work.
* atalay.demiray@yale.edu

## Abstract

Abruption and retention, two types of abnormal placental separation, are associated with significant morbidity and mortality. Though advancements in obstetric management have improved peripartum injury, patients who experience abnormal placental separation may be at risk for long-term complications. This study evaluates long-term maternal mortality in patients with abruption or retention compared to those with normal placental separation. In our cohort of 638,911 vaginal deliveries (625,890 normal, 5,435 abruption, 7,586 retention), the mortality rate was 6.4 per 1,000 in normal deliveries, 9.8 per 1,000 in abruption, and 12.0 per 1,000 in retention. When controlling for demographic factors (age, race, social determinants of health), placental retention was associated with a 95% increased mortality risk (HR 1.95, 95% CI 1.59–2.40, p<0.001) and placental abruption with a 59% increased risk (HR 1.59, 95% CI 1.21–2.08, p<0.001). After excluding deaths within 42 days, the association with retention remained significant (HR 1.93, p<0.001) while abruption lost significance (HR 1.31, p=0.089), suggesting abruption-associated mortality may be driven by acute complications. Piecewise Cox regression confirmed these temporal patterns, with retention showing persistently elevated risk across all follow-up periods. A variety of health outcomes were associated with either abruption, retention or in both abnormal placental separation groups. More research is needed to understand the mechanisms associated with abnormal placental separation and contributors to long-term mortality.

## Introduction

Two primary complications of placental separation are abruption and retention. Placental abruption involves the complete or partial separation of a normally implanted

**Data availability statement:** The data underlying this study were accessed through the TriNetX global federated research network (https://trinetx.com), a third-party commercial platform that provides access to de-identified electronic health record data from participating healthcare organizations. The authors accessed these data under an institutional license governed by a data use agreement with TriNetX. Because the data are proprietary and subject to contractual licensing restrictions, the authors are legally prohibited from publicly sharing, redistributing, or depositing the data in a public repository. The authors did not receive any special access privileges beyond those available to authorized TriNetX users. This study was reviewed by the Penn State Human Subjects Protection Office and determined to not meet the definition of human subject research (Study ID: STUDY00020794). Qualified researchers may request access to TriNetX data by contacting TriNetX directly (email: join@trinetx.com), subject to approval, execution of a data use agreement, and any applicable fees. Inquiries regarding data access or ethical oversight may also be directed to the Penn State Office for Research Protections, Human Research Protection Program (phone: 814-865-1775; email: irb-orp@psu.edu; website: research.psu.edu/irb).

**Funding:** The author(s) received no specific funding for this work.

**Competing interests:** The authors have declared that no competing interests exist.

placenta before delivery. Placental retention happens when the placenta does not fully separate within 30–60 minutes after birth. Both conditions pose significant risks. Abruption can lead to severe complications such as hemorrhage, need for transfusions, disseminated intravascular coagulopathy (DIC), renal failure, and even mortality [1–5]. Similarly, placental retention can result in hemorrhage, invasive procedures, endomyometritis and mortality [6–8].

While advancements in obstetric care have significantly reduced placental separation associated mortality [3,9–11], individuals with placental separation complications may still be at risk for long-term health issues. Research indicates that these two conditions can heighten the risk of cardiovascular, neurological and oncological problems [12–14]. Specifically, patients who experience abruption have an elevated long-term risk of cardiovascular and cerebrovascular diseases [1,12,15–21], while those who experience placental retention accompanied by hemorrhage face increased risks of cardiovascular disease and cancer [13]. These findings highlight the need for ongoing monitoring and preventative care for individuals who have experienced these complications, emphasizing the importance of early recognition and intervention to mitigate long-term health risks.

The reasons behind these associations remain unclear [22–25]. While biological factors likely play a role, the clinical approach to post-delivery management of patients who experience abnormal placental separation may also contribute. The absence of standardized protocols for postpartum follow-up may lead to missed or delayed diagnoses and/or treatment, disproportionately affecting some populations. This uncertainty is also complicated by limited mechanistic understanding of the etiology and outcomes of placental dysfunction. The tendency to examine abnormal placental separation into their separate clinical diagnoses, abruption or retention, may impede the identification of molecular pathways that underlby placental dysfunction perinatally and beyond. Placental dysfunction can include conditions such as abruption, preeclampsia, fetal growth restriction or preterm delivery and are associated with long-term cardiovascular diseases [25,26]. Maternal vascular malperfusion has been found in all these conditions [26] and may be one etiologic component of placental dysfunction. Though retained placenta is not traditionally categorized under placental dysfunction, similarities such as maternal vascular malperfusion [27,28] and associated long-term outcomes [13,14] suggests mechanistic similarities. Categorizing abruption and retention under the broader concept of abnormal placental separation may improve our understanding of the pathogenesis and inform both acute and long-term care strategies.

The primary objective of this study is to compare long-term mortality in patients with pathologic placental separation in an index delivery. We hypothesize that patients with either type of abnormal placental separation, abruption or retention, will have a higher risk of long-term mortality as compared to normally separating placentas. We chose to focus on a clear outcome such as mortality as this is clinically significant and accurately represented in electronic health records (EHR). The secondary objective of this study is to analyze associations between abruption and retention with health outcomes to further elucidate potential disease mechanisms and risk factors.

## Materials and methods

### Study design

A retrospective observational case control cohort study was conducted using the TriNetX EHR database of all women who delivered vaginally between the ages of 15–54 with a reported inpatient encounter. TriNetX is a global federal health research network that provides researchers access to continuously updated data elements from participating health care organizations, predominantly in the United States. TriNetX is certified to the ISO 27001:2013 standard and protects health-care data by maintaining compliance with the Health Insurance Portability and Accountability Act (HIPPA) Security Rule. The EHR data elements are aggregated, de-identified, and include demographic characteristics, diagnoses, procedures, medications, laboratory values and genomics, all in compliance with the de-identification standard outlined in Section 164.51(a) of the HIPPA privacy rule. As no protected health information is received by the user, we were provided a waiver from the Penn State Health Institutional Review Board to perform this study (STUDY00020794). Study design, conduct and result reporting were constructed using the Strengthening the Reporting of Observational Studies in Epidemiology (STROBE) guidelines.

### Data collection

The dataset was generated and downloaded on 23/02/2024. All deliveries were identified in TriNetX on April 5, 2024, using related common procedural terminology (CPT) codes for vaginal deliveries and cesarean section (N = 1,483,438). The earliest delivery date was identified for all patients and used to determine mode of delivery (i.e., vaginal or cesarean section). Deliveries complicated by abnormal placental separation were identified using International Classification of Diseases (ICD) diagnostic codes for placental abruption and retained placenta. Placental abruption was defined using ICD-9-CM codes 641.2x and ICD-10-CM codes O45.x, while retained placenta was defined using ICD-9-CM codes 667.0x–667.1x and ICD-10-CM codes O72.0–O72.2 (full code lists provided in S1 Table). Dates for placental separation ICD codes were identified in order to ensure that placental separation status co-occurred with the identified delivery for each patient. Placental separation ICD codes 6 weeks before delivery and up to 6 weeks after delivery were used to assign a delivery to a placental outcome group. Deliveries that did not have co-occurring placental separation ICD codes were excluded in the primary analysis but were retained for the purposes of sensitivity analysis for mortality. Patients without any ICD codes for placental separation were assigned to the normal delivery group.

We then extracted the earliest and latest encounter dates for each patient. Patients with less than 30 days of follow-up, with records indicating both abruption and retention, and those delivered by cesarean section were excluded to ensure mutually exclusive groupings and to control for the possible impact of delivery mode on maternal morbidity and mortality. The final dataset comprised of 638,911 patients after applying these inclusion and exclusion criteria. We then obtained the following data: age, race, ethnicity, Centers for Disease Control and Prevention (CDC) classification- related maternal morbidity diagnoses, CPT codes, healthcare common procedure coding system (HCPS) codes, and relevant health outcomes using ICD codes [29,30]. In order to control for possible differences in medical record data entry, we excluded deliveries prior to January 1, 2008, as EHR use prior to 2008 was inconsistent (Fig 1). The primary outcome was all-cause mortality following the index delivery. We examined mortality across the full follow-up period as well as within specific time windows to distinguish short-term from long-term mortality patterns.

### Statistical analysis

Categorical variables, including demographic factors and health conditions, were summarized as counts and percentages. Comparisons between groups were performed using the Chi-square test. Continuous variables, such as maternal year of birth and follow-up duration, were summarized as means and standard deviations. Since continuous variables were not normally distributed (assessed visually via histograms and confirmed by Shapiro-Wilk tests), comparisons across groups were performed using the Kruskal-Wallis test, a non-parametric method appropriate for non-normally distributed data.

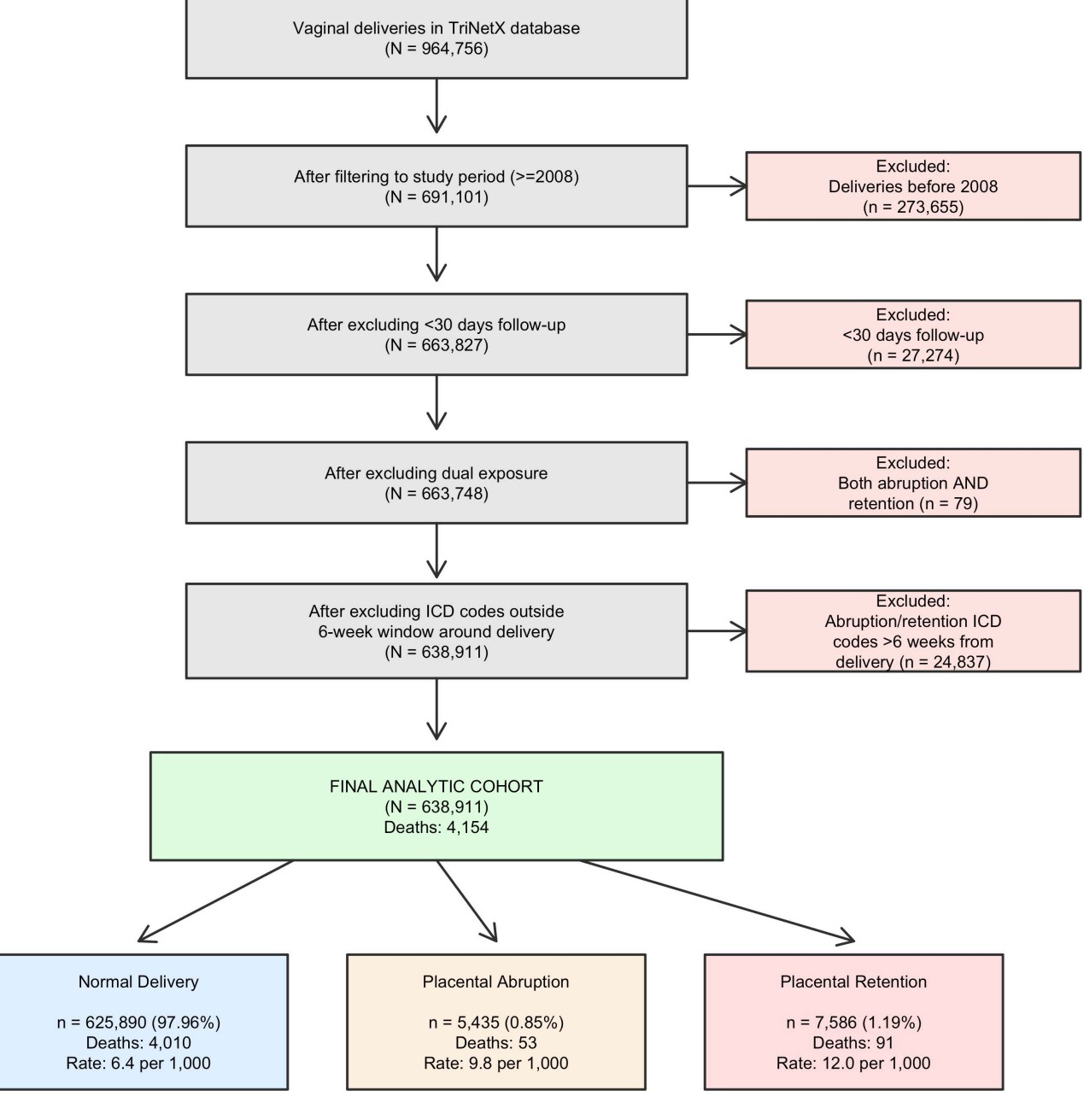

ICD codes within 6 weeks before or after delivery date used to define placental separation groups

**Fig 1. Data Collection Flowchart.**

Cox proportional hazards regression models were used to estimate hazard ratios (HRs) and 95% confidence intervals (CIs) for the association between placental outcomes and subsequent mortality. The hazard ratio presents the relative risk of death at any given time point, comparing the exposed group (abruption or retention) to the reference group (normal delivery). We employed a directed acyclic graph (DAG)-informed approach to covariate selection, distinguishing between true confounders and potential mediators or colliders. Our primary adjusted model controlled for demographic confounders: maternal age at delivery, race, and social determinants of health (SDOH) indicator. SDOH were defined using ICD-10-CM–based indicators available within the TriNetX platform and were analyzed as binary variables indicating the presence or absence of documented social risk indicators. We deliberately avoided adjusting for health conditions that could be consequences of abnormal placental separation or acute delivery complications such as DIC, hemorrhage. A secondary model additionally adjusted for pre-existing conditions (diabetes, hypertension, obesity, chronic kidney disease) to assess the robustness of associations when accounting for baseline health status.

To understand the temporal pattern of mortality risk, we conducted multiple complementary analyses (Table 2): (1) Primary analysis: Full follow-up period with all 4,154 deaths included to provide overall mortality risk estimates. (2) Short-term mortality: Follow-up censored at 365 days, capturing only the 933 deaths occurring within the first year after delivery. (3) Excluding acute postpartum deaths: Patients who died within 42 days of delivery (n = 346) were removed from the analysis to assess whether associations persist beyond the immediate postpartum period. (4) Long-term mortality only: Patients who died within one year of delivery (n = 933) were removed to isolate long-term mortality risk. Additionally, to examine how associations varied across follow-up time, we conducted piecewise Cox regression analyses by fitting separate models for distinct time intervals (0–1, 1–2, 2–5, and >5 years). For each interval, patients remaining at risk at the interval start were included, with events after the interval endpoint censored (S4 Table).

For each Cox proportional hazards model, the proportional hazards assumption was evaluated both graphically using Schoenfeld residuals and formally tested using the global and variable-specific test. Variable-specific testing identified which individual covariates contributed to PH violations because global PH violations driven by adjustment covariates do not invalidate hazard ratio estimates for the primary exposure variable if that variable itself satisfies the assumption (S3 Table). Multiple comparison corrections were applied using Bonferroni and Benjamini-Hochberg false discovery rate (FDR) methods for primary mortality outcomes (S5 Table).

Kaplan-Meier survival curves were generated to visualize survival probabilities across groups, and differences in survival distributions were formally tested using the log-rank test. In addition to mortality, associations between placental outcomes and a range of diagnosed health conditions (including hypertension, cardiovascular disease, sepsis, among others) were investigated. For these health outcome associations, multivariable logistic regression models were used to estimate adjusted odds ratios (ORs) with 95% CIs, as these conditions were assessed cross-sectionally rather than as time-to-event outcomes. Models were adjusted for maternal age, race, and SDOH indicator. Covariates included in multivariable models were selected based on theoretical relevance and data availability. No stepwise selection methods were applied. Multicollinearity among covariates was assessed using variance inflation factors (VIFs), and no VIF exceeded the threshold of 5.

HRs for mortality outcomes and ORs for health condition associations with 95% CIs and corresponding p-values are reported. All hypothesis tests were two-sided, and a p-value < 0.05 was considered statistically significant. No corrections for multiple comparisons were applied due to the exploratory nature of secondary analyses, though findings are interpreted cautiously. All statistical analyses were conducted using R software, version 4.5.2 (R Foundation for Statistical Computing, Vienna, Austria). The complete R code used to perform the analyses is available (code in S7 Appendix). We cannot share the raw data file because it is licensed by TriNetX and their data use policies do no permit redistribution.

## Results

Our final cohort comprised 638,911 vaginal deliveries: 625,890 (97.96%) with normal placental separation, 5,435 (0.85%) with placental abruption, and 7,586 (1.19%) with placental retention. Baseline characteristics are presented in Table 1.

**Table 1. Baseline characteristics of the study population by placental separation type.**

| | Normal Delivery | Placental Abruption | Placental Retention | P-value |
|---|---|---|---|---|
| n | 625,890 | 5,435 | 7,586 | |
| Deaths, n (rate per 1,000) | 4,010 (6.4) | 53 (9.8) | 91 (12) | |
| Person-years of follow-up | 2,171,019 | 18,074 | 24,832 | |
| Mortality rate per 1,000 person-years | 1.85 | 2.93 | 3.66 | |
| Follow-up time, mean (SD), years | 3.47 (3.27) | 3.33 (3.11) | 3.27 (3.01) | |
| Maternal age at delivery, mean (SD) | 28.4 (6) | 29 (6.1) | 29.8 (6.1) | |
| Year of birth, mean (SD) | 1989.53 (6.61) | 1988.91 (6.64) | 1988.51 (6.61) | |
| **Race, n (%)** | | | | <0.001 |
| White | 373,373 (59.65%) | 2,997 (55.14%) | 4,631 (61.05%) | |
| Black or African American | 89,242 (14.26%) | 849 (15.62%) | 885 (11.67%) | |
| Asian | 29,361 (4.69%) | 317 (5.83%) | 343 (4.52%) | |
| Native Hawaiian or Other Pacific Islander | 2,029 (0.32%) | 26 (0.48%) | 27 (0.36%) | |
| American Indian or Alaska Native | 1,954 (0.31%) | 16 (0.29%) | 37 (0.49%) | |
| Other Race | 41,350 (6.61%) | 399 (7.34%) | 553 (7.29%) | |
| Unknown | 88,581 (14.15%) | 831 (15.29%) | 1,110 (14.63%) | |
| **Ethnicity, n (%)** | | | | <0.001 |
| Hispanic or Latino | 162,521 (25.97%) | 1,200 (22.08%) | 1,714 (22.59%) | |
| Not Hispanic or Latino | 361,902 (57.82%) | 3,291 (60.55%) | 4,590 (60.51%) | |
| Unknown | 101,467 (16.21%) | 944 (17.37%) | 1,282 (16.90%) | |
| **SOCIOECONOMIC FACTORS, n (rate per 1,000)** | | | | |
| Social determinants of health indicator | 47,315 (75.6) | 528 (97.1) | 590 (77.8) | <0.001 |
| **Marital Status, n (%)** | | | | <0.001 |
| Married | 134,549 (21.50%) | 1,108 (20.39%) | 1,404 (18.51%) | |
| Single | 125,403 (20.04%) | 1,267 (23.31%) | 1,285 (16.94%) | |
| Unknown | 365,938 (58.47%) | 3,060 (56.30%) | 4,897 (64.55%) | |
| **RISK FACTORS, n (rate per 1,000)** | | | | |
| Alcohol use | 9,897 (15.8) | 146 (26.9) | 173 (22.8) | <0.001 |
| Tobacco use | 65,635 (104.9) | 917 (168.7) | 932 (122.9) | <0.001 |
| Substance use | 46,954 (75) | 727 (133.8) | 694 (91.5) | <0.001 |
| **HEALTH CONDITIONS** | | | | |
| Diabetes | 18,120 (29) | 203 (37.4) | 297 (39.2) | <0.001 |
| Hypertension | 46,978 (75.1) | 533 (98.1) | 799 (105.3) | <0.001 |
| Obesity | 147,539 (235.7) | 1,219 (224.3) | 2,036 (268.4) | <0.001 |
| Chronic kidney disease | 7,982 (12.8) | 98 (18) | 157 (20.7) | <0.001 |
| Asthma | 71,383 (114.1) | 743 (136.7) | 1,064 (140.3) | <0.001 |
| Mental health disorder | 188,048 (300.4) | 1,970 (362.5) | 2,871 (378.5) | <0.001 |
| **PREGNANCY & DELIVERY** | | | | |
| Multiple gestation | 14,062 (22.5) | 212 (39) | 315 (41.5) | <0.001 |
| Placenta previa | 33,429 (53.4) | 482 (88.7) | 593 (78.2) | <0.001 |
| Preeclampsia | 40,718 (65.1) | 517 (95.1) | 722 (95.2) | <0.001 |
| Eclampsia | 1,528 (2.4) | 35 (6.4) | 42 (5.5) | <0.001 |
| Postpartum hemorrhage | 46,592 (74.4) | 786 (144.6) | 2,791 (367.9) | <0.001 |
| DIC | 4,148 (6.6) | 125 (23) | 96 (12.7) | <0.001 |
| **PROCEDURES** | | | | |
| Hysterectomy | 682 (1.1) | 7 (1.3) | 42 (5.5) | <0.001 |
| Blood transfusion | 11,548 (18.5) | 331 (60.9) | 585 (77.1) | <0.001 |

Values are n (%) for categorical variables or mean (SD) for continuous variables. Rates per 1,000 calculated as (n/total group n) x 1,000. P-values from chi-square tests for categorical variables and ANOVA for continuous variables.

Health conditions including diabetes, hypertension, and chronic kidney disease were more prevalent in both placental complication groups. The mean follow-up was 3.5 years (SD 3.3), accumulating 2,213,925 total person-years of observation. During follow-up, 4,154 deaths occurred (0.65% overall mortality): 4,010 in the normal group (mortality rate: 6.4 per 1,000 deliveries or 1.85 per 1,000 person-years), 53 in the abruption group (9.8 per 1,000 deliveries or 2.93 per 1,000 person-years), and 91 in the retention group (12.0 per 1,000 deliveries or 3.66 per 1,000 person-years).

In unadjusted Cox proportional hazard analyses, both placental abruption (HR 1.62, 95% CI 1.23–2.12, p < 0.001) and placental retention (HR 2.04, 95% CI 1.66–2.51, p < 0.001) were associated with significantly elevated mortality compared to normal delivery (Table 2). After adjustment for demographic confounders (maternal age, race, SDOH indicator), these associations remained significant: abruption HR 1.59 (95% CI 1.21–2.08, p < 0.001) and retention HR 1.95 (95% CI 1.59–2.40, p < 0.001). Further adjustment for health conditions (diabetes, hypertension, obesity, and chronic kidney disease) modestly attenuated the associations: abruption HR 1.56 (95% CI 1.19–2.04, p = 0.001) and retention HR 1.87 (95% CI 1.52–2.30, p < 0.001).

To understand the temporal pattern of mortality risk, we conducted complementary sensitivity analyses (S6 Table). Analysis of short-term mortality, with follow-up censored at 365 days, revealed elevated mortality risk for both conditions: abruption HR 1.88 (95% CI 1.13–3.13, p = 0.016) and retention HR 1.75 (95% CI 1.12–2.73, p = 0.013). This indicates that both placental complications are associated with significantly increased mortality risk in the post-delivery period.

When excluding deaths within 42 days of delivery, the retention association remained robust (HR 1.93, 95% CI 1.55–2.41, p < 0.001), while the abruption association was attenuated and no longer statistically significant (HR 1.31, 95% CI 0.96–1.79, p = 0.089). This suggests that a substantial portion of abruption-associated mortality occurs within the acute postpartum period.

Analysis excluding all deaths within the first year showed persistent mortality associations for both conditions: abruption HR 1.49 (95% CI 1.08–2.06, p = 0.014) and retention HR 2.02 (95% CI 1.59–2.55, p < 0.001). These findings indicate that both placental complications are associated with elevated long-term mortality risk that extends well beyond the traditional postpartum surveillance period, with retention showing particularly strong and consistent associations across all time horizons.

The proportional hazards assumption was evaluated using Schoenfeld residuals with both global and variable-specific tests (S3 Table). While the global test indicated a violation (p = 0.0002), variable-specific testing revealed that the primary exposure variable (placental separation group) satisfied the proportional hazards assumption (p = 0.62). The global violation was driven by adjustment covariates (maternal age p = 0.011; race p = 0.0004), which does not invalidate the hazard

**Table 2. Hazard ratios for all-cause mortality by placental separation type.**

| | Placental Abruption | | Placental Retention | |
|---|---|---|---|---|
| | HR (95% CI) | P-value | HR (95% CI) | P-value |
| Unadjusted | 1.62 (1.23-2.12) | <0.001 | 2.04 (1.66-2.51) | <0.001 |
| Adjusted (Demographics)[a] | 1.59 (1.21-2.08) | <0.001 | 1.95 (1.59-2.40) | <0.001 |
| + Pre-existing conditions[b] | 1.56 (1.19-2.04) | 0.001 | 1.87 (1.52-2.30) | <0.001 |
| Short-term (within 1 year)[c] | 1.88 (1.13-3.13) | 0.016 | 1.75 (1.12-2.73) | 0.013 |
| Excluding deaths ≤42 days[d] | 1.31 (0.96-1.79) | 0.089 | 1.93 (1.55-2.41) | <0.001 |
| Long-term only (excluding ≤1 year)[e] | 1.49 (1.08-2.06) | 0.014 | 2.02 (1.59-2.55) | <0.001 |

[a]Adjusted for maternal age, race/ethnicity, and social determinants of health (SDOH) indicator.

[b]Additionally adjusted for diabetes, hypertension, obesity, and chronic kidney disease.

[c]Follow-up censored at 365 days; only deaths within first year counted.

[d]Patients who died within 42 days (WHO maternal mortality window) excluded from analysis.

[e]Patients who died within 1 year excluded; assesses long-term mortality only.

ratio estimates for the primary exposure. The hazard ratios reported should be interpreted as average effects across the follow-up period.

Exploratory secondary analyses examined the association between placental complications and health outcomes (Table 3). Health outcomes analyzed included CDC-related maternal morbidity diagnoses, related CPT, HCPS codes, and predefined ICD codes [29,30]. Both placental abruption and retention were associated with significantly elevated odds of hypertension, diabetes, chronic kidney disease, lupus, asthma, heart failure, obesity and mental health disorders compared to normal delivery. These associations remained significant after adjustment for demographic factors. Pregnancy-specific conditions that were significantly increased in both abruption and retention included: placenta previa, preeclampsia, eclampsia, multiple gestation, PPH. Conditions that were significantly elevated in abruption only included: ischemic heart disease, and puerperal cerebrovascular disease. Conditions that were significantly elevated in retention only included: gestational hypertension, cardiovascular disease, hysterectomy, pulmonary hypertension and thyroid cancer.

## Discussion

Our primary aim was to compare long-term all-cause maternal mortality in those with pathologic placental separation to those with normal separation. Our results demonstrate that both placental abruption and placental retention are associated with significantly increased long-term maternal mortality compared to normal placental separation. Additionally, our sensitivity analyses reveal distinct temporal patterns of mortality risk between these two conditions. Placental retention showed consistently higher mortality across all analyses: short-term within the first year (HR 1.75), after excluding 42-day deaths (HR 1.93), and in long-term only analysis (HR 2.02), suggesting persistent pathophysiological processes that continue to elevate mortality risk years after delivery. In contrast, placental abruption showed the highest relative risk in the short-term (HR 1.88), marked attenuation after excluding acute postpartum deaths (HR 1.31, p = 0.089), but a return to significance in long-term only analysis (HR 1.49, p = 0.014).

These distinct patterns highlight the importance of looking beyond the traditional postpartum window. Literature on maternal mortality within the first 42 days reflects high-acuity causes like postpartum hemorrhage, embolism, postpartum eclampsia, and infection. However, a large portion of maternal deaths occur after 42 days from delivery and reflect a shift to conditions like peripartum cardiomyopathy, other cardiovascular diseases, and chronic non-cardiovascular conditions. Maternal mortality reviews typically analyze mortality events during pregnancy up to one year after delivery and organize etiologies into broad categories (e.g., obstetric hemorrhage) [31–35]. This broad approach, however, can obfuscate the mortality patterns of each individual pathology. By comparing abruption and retention together, our results suggest that while both are associated with acute morbidity, their long-term trajectories may diverge.

Both conditions are known to be associated with serious obstetric hemorrhage resulting in acute maternal morbidity and mortality [1–8,15,36–41]. Consistent with the literature, our results show associations with increased health interventions such as critical care use, transfusion, and mechanical ventilation in both groups [1–8,15,36–41]. The exact chronicity of these interventions in relation to the abnormal placental separation event, however, cannot be determined from our data set. Little is known about the long-term implications for those who survive abnormal placental separation events. There is a growing body of literature suggesting connections between long-term maternal morbidity and mortality and conditions that occurred during pregnancy [31,42,43]. Abruption is associated with long-term maternal mortality and morbidity from both cardiovascular and non-cardiovascular etiologies [15–21]. Similarly, retention with hemorrhage may be associated with long-term cardiovascular and cancer risks [13,14].

We identified distinct clusters of health conditions associated with each separation type that may offer insight into these differing mortality risks. In the abruption group, we found a significantly increased prevalence of ischemic heart disease and puerperal cerebrovascular accidents. Both conditions are associated with maternal mortality during delivery or in the immediate postpartum timeframe [44–47]. Conversely, in the retention group, we found a significantly increased prevalence of cardiovascular disease, pulmonary hypertension, and thyroid cancer. These conditions may be more insidious

**Table 3. Significant clinical associations with pathologic placental separation states after adjusting for demographic factors.**

| Condition | Placental Retention (adj OR) | Placental Abruption (adj OR) | Comparative Directionality |
|---|---|---|---|
| **Abnormal Placentation** | | | |
| Placenta previa | 1.44 [1.32, 1.56] p=<0.001 | 1.70 [1.55, 1.87] p=<0.001 | Similar |
| **Obstetric** | | | |
| Preeclampsia | 1.53 [1.41, 1.65] p=<0.001 | 1.50 [1.37, 1.65] p=<0.001 | Similar |
| Eclampsia | 2.32 [1.70, 3.15] p=<0.001 | 2.56 [1.83, 3.58] p=<0.001 | Similar |
| Gestational hypertension | 1.23 [1.14, 1.32] p=<0.001 | 1.07 [0.98, 1.17] p=0.119 | Different (sig. R only) |
| Multiple gestation | 1.84 [1.64, 2.07] p=<0.001 | 1.73 [1.50, 1.98] p=<0.001 | Similar |
| PPH | 7.31 [6.97, 7.67] p=<0.001 | 2.09 [1.94, 2.26] p=<0.001 | Similar |
| **Health Behaviors** | | | |
| Alcohol | 1.50 [1.29, 1.75] p=<0.001 | 1.69 [1.43, 1.99] p=<0.001 | Similar |
| Tobacco | 1.28 [1.20, 1.38] p=<0.001 | 1.78 [1.65, 1.91] p=<0.001 | Similar |
| Substance use | 1.34 [1.24, 1.45] p=<0.001 | 1.95 [1.80, 2.12] p=<0.001 | Similar |
| **Health System** | | | |
| Transfusion | 4.58 [4.20, 4.99] p=<0.001 | 3.40 [3.04, 3.81] p=<0.001 | Similar |
| Mechanical ventilation | 1.88 [1.38, 2.56] p=<0.001 | 3.44 [2.63, 4.49] p=<0.001 | Similar |
| Critical care use | 1.25 [1.08, 1.45] p=0.004 | 1.89 [1.64, 2.18] p=<0.001 | Similar |
| Hysterectomy | 4.42 [3.23, 6.05] p=<0.001 | 1.11 [0.53, 2.35] p=0.778 | Different (sig. R only) |
| **CHRONIC** | | | |
| Hypertension | 1.37 [1.27, 1.48] p=<0.001 | 1.27 [1.16, 1.39] p=<0.001 | Similar |
| Diabetes | 1.26 [1.12, 1.41] p=<0.001 | 1.21 [1.05, 1.39] p=0.009 | Similar |
| Chronic kidney disease | 1.63 [1.39, 1.91] p=<0.001 | 1.41 [1.15, 1.73] p=<0.001 | Similar |
| Mental health | 1.42 [1.35, 1.49] p=<0.001 | 1.33 [1.26, 1.41] p=<0.001 | Similar |
| Lupus | 1.44 [1.07, 1.93] p=0.017 | 1.92 [1.42, 2.60] p=<0.001 | Similar |
| Asthma | 1.31 [1.23, 1.40] p=<0.001 | 1.23 [1.14, 1.33] p=<0.001 | Similar |
| Cardiovascular disease | 1.33 [1.12, 1.59] p=0.001 | 1.12 [0.89, 1.40] p=0.325 | Different (sig. R only) |
| Pulmonary hypertension | 1.62 [1.05, 2.50] p=0.028 | 1.56 [0.94, 2.60] p=0.088 | Different (sig. R only) |
| Thyroid cancer | 1.46 [1.01, 2.13] p=0.046 | 1.03 [0.59, 1.78] p=0.922 | Different (sig. R only) |
| Obesity | 1.19 [1.13, 1.25] p=<0.001 | 0.91 [0.85, 0.97] p=0.004 | Different (sig. R&A) |
| **Acute** | | | |
| Acute respiratory distress syndrome | 1.94 [1.56, 2.40] p=<0.001 | 2.20 [1.74, 2.78] p=<0.001 | Similar |
| Air embolism | 1.67 [1.30, 2.15] p=<0.001 | 1.86 [1.40, 2.47] p=<0.001 | Similar |
| Shock | 1.89 [1.48, 2.43] p=<0.001 | 2.06 [1.56, 2.72] p=<0.001 | Similar |
| Sepsis | 1.78 [1.52, 2.08] p=<0.001 | 1.80 [1.50, 2.15] p=<0.001 | Similar |
| Puerperal cerebrovascular disease | 1.24 [0.96, 1.59] p=0.102 | 1.43 [1.08, 1.89] p=0.013 | Different (sig. A only) |
| **Either Acute or Chronic** | | | |
| Disseminated intravascular coagulopathy | 1.86 [1.52, 2.29] p=<0.001 | 3.42 [2.85, 4.09] p=<0.001 | Similar |
| Heart failure | 1.72 [1.32, 2.24] p=<0.001 | 1.80 [1.33, 2.43] p=<0.001 | Similar |
| Anemia | 1.44 [1.37, 1.51] p=<0.001 | 1.26 [1.18, 1.33] p=<0.001 | Similar |
| Ischemic heart disease | 0.96 [0.68, 1.36] p=0.819 | 1.53 [1.10, 2.12] p=0.011 | Different (sig. A only) |

R=Retention, A=Abruption.

All conditions in the 'Similar' column were significantly elevated in both abruption and retention groups after adjustment. All conditions in the 'Different' column were significantly elevated in either abruption or retention groups.

Note: 'Similar directionality' indicates both conditions show significantly elevated risk (HR>1, p<0.05) for the outcome, regardless of magnitude difference. 'Different' indicates only one condition shows significant association.

 

compared to the acute vascular events seen in the abruption group, potentially explaining why the elevated mortality risk in retention persists well beyond the immediate postpartum period. Our data is unable to distinguish the timing of association as these health conditions may have existed prior to or occurred after the abnormal placental separation event. Thus, additional studies are needed to further clarify this.

Though placental dysfunction and cardiovascular disease have mechanistic similarities [1,12,37,42], how these mechanisms may contribute to long-term morbidity and mortality is currently unknown. Answering these questions is clinically relevant in guiding both postpartum care and the obstetric to primary care transition. While our findings are observational, they underscore the need for further research to inform clinical recommendations for postpartum management, specifically regarding the timing, frequency, and necessary interventions for long-term follow-up in patients who experience abnormal placental separation.

The secondary aim of this study was to explore possible disease mechanisms associated with abnormal placental separation. We found a significantly increased prevalence of placenta previa and preeclampsia/eclampsia in abruption and retention groups. Further investigation into mechanisms involving implantation, placental migration, matrix metalloproteinases, apoptosis, inflammatory cytokine expression, and uterine/endometrial blood supply is needed to validate the exploratory association we identified between abnormal placental separation and placenta previa [38,39,48,49]. Additionally, further translational research on angiogenesis-related biomarkers such as soluble fms-like tyrosine kinase-1 (sFlt-1), placental growth factor (PlGF), and the sFlt-1/PlGF ratio is necessary to substantiate the association we found between abnormal placental separation and preeclampsia/eclampsia [36,46,50–53].

Though our results cannot identify specific biologic mechanisms, inflammation may underlie the associations we noted between abnormal placental separation and the variety of health conditions examined in our study [27,28,54–62]. We observed significant associations between hypertension, diabetes, chronic kidney disease, mental health conditions, lupus, asthma, sepsis, heart failure, and acute respiratory distress with both abnormal placental separation groups. The exact molecular mechanisms that underlie abruption and retention are currently unknown, but hypothesized mechanisms demonstrate striking similarities involving placental hypoperfusion, dysfunctional implantation/placentation, and the effects of mechanical forces on the placenta [1,6,8,63]. The significant similarities demonstrated in our study between associated health conditions with abruption and retention lends credence to the notion of possible shared molecular mechanisms in pathophysiology. Elucidating the interaction between immune function and abnormal placental separation is a promising area for future study.

The strengths of this study include our large multi-center study cohort, case control study design, comprehensive health conditions list, and our comparative approach analyzing both abruption and retention. We utilized a cohort of over 638,000 vaginal deliveries with more than 2.2 million person-years of follow-up (mean 3.5 years, SD 3.3), providing substantial statistical power to detect mortality differences. We focused on the clear outcome of mortality and excluded deliveries prior to 2008 to control for possible EHR data quality issues. Our Cox regression models had adequate statistical power with events-per-variable ratios exceeding 400 (S2 Table). The proportional hazards assumption was evaluated using Schoenfeld residuals with both global and variable-specific tests (S3 Table). While the global test indicated violation (p = 0.0002), variable-specific testing revealed that the primary exposure variable (placental separation group) satisfied the proportional hazards assumption (p = 0.62). The global violation was driven by adjustment covariates (maternal age p = 0.011; race p = 0.0004), which does not invalidate the hazard ratio estimates for the primary exposure. The hazard ratios reported should be interpreted as average effects across the follow-up period. Further sensitivity analyses confirmed the robustness of our primary findings.

There are several limitations to this study. Due to database limitations, diagnostic codes were used to categorize placental separation groups and identify all health conditions. Thus, our study may be subject to misclassification bias. Statistically, this misclassification may increase a type II error and possibly miss some significant associations. For example, one study reported on the differences in retained placenta prevalence when using ICD codes (2.8%) versus clinical review

(7%) [64]. Despite the limitations of using a large scale database, using TriNetX may improve recognition of maternal deaths compared to standard practice of requesting maternal mortality data through medical informatics [65]. TriNetX utilization can facilitate rapid cohort identification, generating real-world evidence through regularly up-dated information and enabling large-scale observational data [66]. However, it is important to note that our study design does not allow for causational or chronologic associations due to the limitations of TriNetX. The elevated health conditions associated with abnormal placental separation could reflect associations either prior to or following the index delivery. Thus, they should be considered to demonstrate relationships with possible underlying mechanisms that need to be explored and validated in future studies. Finally, we could not fully account for the role that obesity may have played. Our results showed obesity to be elevated in retention and reduced in abruption consistent with the literature [41,50,63,67,68], however, we could not fully control for this covariate due to inconsistent information on BMI or pregnancy weight gain in the TriNetX dataset. This remains a limitation, as obesity is a known confounder for both placental complications and long-term cardiovascular mortality. Without information on BMI, we were only able to account for obesity through ICD codes which may have been inconsistent and lead to mis-classification bias. This potential bias may have mediated the mortality patterns and health condition associations that we noted with abruption and retention. Future studies that can fully account for BMI are needed to build upon our findings.

## Conclusions

Long-term mortality risk in patients who experience abruption or retention during an index delivery is elevated when adjusting for demographic factors. However, the temporal nature of this risk varies: short-term mortality events appear to drive the risk in patients with abruption, whereas retention is associated with a persistent elevation in mortality extending years beyond delivery. Comparing and contrasting significant health associations between abruption and retention groups can provide exploratory information to guide further analyses. More research is needed to identify the mechanistic contributors to long-term mortality in those with abnormal placental separation.

## Supporting information

**S1 Table. ICD-9-CM and ICD-10-CM codes used for placental abruption, placental retention, and related conditions.**
(DOCX)

**S2 Table. Events per variable (EPV) analysis for Cox regression models demonstrating adequate statistical power.**
(CSV)

**S3 Table. Proportional hazards assumption tests using Schoenfeld residuals with variable-specific analysis.**
(CSV)

**S4 Table. Time-specific hazard ratios from piecewise Cox regression analyses by follow-up period.**
(CSV)

**S5 Table. Multiple comparison corrections using Bonferroni and Benjamini-Hochberg false discovery rate (FDR) methods for primary mortality comparisons.**
(CSV)

**S6 Table. Complete sensitivity analysis results for mortality models including short-term (within 1 year), excluding acute deaths (≤42 days), and long-term only (excluding ≤1 year) analyses.**
(CSV)

**S7 Appendix. R code for statistical analysis.**
(R)

## Author contributions

**Conceptualization:** Sona Jasani, Atalay Demiray, Conrad Krawiec.

**Data curation:** Sona Jasani, Atalay Demiray, Julia Stevenson, Conrad Krawiec.

**Formal analysis:** Sona Jasani, Atalay Demiray, Conrad Krawiec.

**Investigation:** Julia Stevenson, Conrad Krawiec.

**Methodology:** Sona Jasani, Atalay Demiray, Conrad Krawiec.

**Project administration:** Sona Jasani.

**Supervision:** Sona Jasani, Conrad Krawiec.

**Validation:** Sona Jasani, Conrad Krawiec.

**Visualization:** Atalay Demiray.

**Writing – original draft:** Sona Jasani, Atalay Demiray, Julia Stevenson, Conrad Krawiec.

**Writing – review & editing:** Sona Jasani, Atalay Demiray, Julia Stevenson, Conrad Krawiec.

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
