## [Decision Letter · Decision Letter 0]

31 Jan 2026

A comparison of long-term mortality associated with pathologic placental separation: highlighting possible trends and mechanisms

PLOS One

Dear Dr. Demiray,

Thank you for submitting your manuscript to PLOS ONE. After careful consideration, we feel that it has merit but does not fully meet PLOS ONE’s publication criteria as it currently stands. Therefore, we invite you to submit a revised version of the manuscript that addresses the points raised during the review process.

The manuscript addresses an important and clinically relevant topic, but the peer reviewers have raised several substantive concerns that require careful attention before the work can be reconsidered. In particular, the revision should focus closely on the reviewers’ comments regarding clarity and structure of the tables, definition and classification of outcomes and covariates, justification of the adjustment strategy, assessment of model assumptions, and ensuring that the interpretation and conclusions are fully supported by the analyses. A thorough, point-by-point response to all reviewer comments is required.

We look forward to receiving your revised manuscript.

Kind regards,

Melvin Marzan, BSc, MSc TM, PhD

Academic Editor

PLOS One

**Journal Requirements:**

2. For studies involving third-party data, we encourage authors to share any data specific to their analyses that they can legally distribute. PLOS recognizes, however, that authors may be using third-party data they do not have the rights to share. When third-party data cannot be publicly shared, authors must provide all information necessary for interested researchers to apply to gain access to the data. (https://journals.plos.org/plosone/s/data-availability#loc-acceptable-data-access-restrictions)

3. Please include captions for your Supporting Information files at the end of your manuscript, and update any in-text citations to match accordingly. Please see our Supporting Information guidelines for more information: http://journals.plos.org/plosone/s/supporting-information....

4. Please upload a new copy of S4_Appendix.tiff as the detail is not clear. Please follow the link for more information:  https://journals.plos.org/plosone/s/figures

Reviewers' comments:

Reviewer's Responses to Questions

**Comments to the Author**

1. Is the manuscript technically sound, and do the data support the conclusions?

Reviewer #1: Partly

Reviewer #2: Yes

Reviewer #3: Yes

Reviewer #4: No

2. Has the statistical analysis been performed appropriately and rigorously?

Reviewer #1: I Don't Know

Reviewer #2: Yes

Reviewer #3: Yes

Reviewer #4: No

3. Have the authors made all data underlying the findings in their manuscript fully available?

Reviewer #1: Yes

Reviewer #2: Yes

Reviewer #3: Yes

Reviewer #4: No

4. Is the manuscript presented in an intelligible fashion and written in standard English?

Reviewer #1: No

Reviewer #2: Yes

Reviewer #3: Yes

Reviewer #4: Yes

Reviewer #1: This manuscript addresses an important topic: the long-term impact in women after experiencing pathologic separation of the placenta after vaginal birth. While the topic should be of interest to readers, the presentation of the copious amount of data within the manuscript’s 3 tables is difficult to understand.

Table 1 presents demographic data regarding the study population. That table shows that placental retention and placental abruption occur in few deliveries. In the line under that entry, under “p” is the value >0.001. It is not clear if that value refers to Number of Deliveries or the Race. Further, I do not understand why this would be subjected to statistical evaluation and which value(s) is different from the referent population. Because the numbers of patients in the 3 groups (Normal, Placental Retention, Placental Abruption) are so dramatically different (Normal = 600,000; the other groups in the range of 5,000 – 7,000) statistics are superfluous. The same issue of disparity in numbers is true for the remaining entries in this table.

Additionally, I question the value of reporting the mean of the years of birth for each group. The means are all within one year of each other. How are they different at p>0.001? A better descriptor would have been maternal age.

SDOH is a qualitative value; the numbers in the cells represent the number of patients categorized as such? How does statistical evaluation of this help in the assessment?

Some entries listed under Health Outcomes are not outcomes. For instance, alcohol use and tobacco use are not health outcomes. Also, procedures carried out later (e.g., Critical Care, Hysterectomy, DIC, Tracheostomy, Transfusion, and Mechanical Ventilation) are not health outcomes. How is Multiple Gestation an outcome of parturition? Once again, when comparing outcomes derived from 600,000 controls, it is not surprising that the values derived from populations of 5,000 – 7,000 would be different. A better comparison would be the rates of occurrence within each population using a method that can evaluate the two aberrant placental groups separately.

Table 2 presents Hazard Ratios for all groups at 30 days and 1 year after parturition. The authors need to specify the formula used to determine the hazard ratios and both how adjustment factors were identified and how they were used to adjust the hazard ratios.

The values in Table 3 are similarly opaque. There are single values in each cell with a p value parenthetically below. When comparing Placental Retention to Placental Abruption, some values are numerically close and described as having similar directionality; but some values are not numerically close, yet considered “similar” (e.g. PPH at 5.46 vs 2.01) whereas some close values are labelled “different” (e.g. Pulmonary Hypertension at 1.55 vs 1.56). The Discussion does little to clarify the data inconsistencies.

Despite my problems with presentation of the data, the story that the authors present is both compelling and interesting. The problems I have is that I do not see that story in the data as theya re presented.

I suggest the authors rework their tables according to the comments above and include a better explanation of how the data were handled. This manuscript addresses an important topic: the long-term impact in women after experiencing pathologic separation of the placenta after vaginal birth. While the topic should be of interest to readers, the presentation of the copious amount of data within the manuscript’s 3 tables is difficult to understand.

Table 1 presents demographic data regarding the study population. That table shows that placental retention and placental abruption occur in few deliveries. In the line under that entry, under “p” is the value >0.001. It is not clear if that value refers to Number of Deliveries or the Race. Further, I do not understand why this would be subjected to statistical evaluation and which value(s) is different from the referent population. Because the numbers of patients in the 3 groups (Normal, Placental Retention, Placental Abruption) are so dramatically different (Normal = 600,000; the other groups in the range of 5,000 – 7,000) statistics are superfluous. The same issue of disparity in numbers is true for the remaining entries in this table.

Additionally, I question the value of reporting the mean of the years of birth for each group. The means are all within one year of each other. How are they different at p>0.001? A better descriptor would have been maternal age.

SDOH is a qualitative value; the numbers in the cells represent the number of patients categorized as such? How does statistical evaluation of this help in the assessment?

Some entries listed under Health Outcomes are not outcomes. For instance, alcohol use and tobacco use are not health outcomes. Also, procedures carried out later (e.g., Critical Care, Hysterectomy, DIC, Tracheostomy, Transfusion, and Mechanical Ventilation) are not health outcomes. How is Multiple Gestation an outcome of parturition? Once again, when comparing outcomes derived from 600,000 controls, it is not surprising that the values derived from populations of 5,000 – 7,000 would be different. A better comparison would be the rates of occurrence within each population using a method that can evaluate the two aberrant placental groups separately.

Table 2 presents Hazard Ratios for all groups at 30 days and 1 year after parturition. The authors need to specify the formula used to determine the hazard ratios and both how adjustment factors were identified and how they were used to adjust the hazard ratios.

The values in Table 3 are similarly opaque. There are single values in each cell with a p value parenthetically below. When comparing Placental Retention to Placental Abruption, some values are numerically close and described as having similar directionality; but some values are not numerically close, yet considered “similar” (e.g. PPH at 5.46 vs 2.01) whereas some close values are labelled “different” (e.g. Pulmonary Hypertension at 1.55 vs 1.56). The Discussion does little to clarify the data inconsistencies.

Despite my problems with presentation of the data, the story that the authors present is both compelling and interesting. The problems I have is that I do not see that story in the data as they are presented. I suggest the authors rework their tables according to the comments above and include a better explanation of how the data were handled.

Reviewer #2: Because some covariates included in the fully adjusted models could represent downstream conditions rather than true con-founders, the authors may wish to clarify the potential for overadjustment and its implications for interpretation

Reviewer #3: In this manuscript, Jasani and colleagues tackle a clinically significant subject by investigating the link between pathologic placental separation and long-term maternal mortality, effectively highlighting the necessity for deeper mechanistic inquiry. The study is methodologically robust and makes a commendable contribution to the existing literature. I have no major objections to the study design or analysis; however, the following minor revisions are recommended to improve clarity:

1- The phrase "long-term mortality" should be revised to "long-term maternal mortality" in both the title and abstract to ensure immediate clarity for the readership.

2- Please explicitly state the specific ICD codes used to define placental abruption (e.g., ICD-9 and ICD-10, specifically the O45 category) in the Methods section. This detail is crucial for study reproducibility.

3- Given that maternal age is a fundamental determinant in reproductive health and obstetric outcomes, the age-related findings warrant a more comprehensive interpretation in the Discussion section. Expanding on this point will help to better contextualize the results and underscore their broader clinical significance.

Reviewer #4: This manuscript addresses an important and understudied topic, long-term mortality following abnormal placental separation (abruption and retained placenta). While the dataset is large and the topic is clinically relevant, the study in its current form has substantial methodological, analytical, and interpretive weaknesses that undermine the validity of its conclusions.

Major Concerns

1. Conceptual and Causal Overreach

The manuscript repeatedly implies that retained placenta itself may contribute causally to long-term mortality, despite explicitly acknowledging that:

• The study design is observational

• Temporality between exposure and many covariates is unclear

• Many “health outcomes” may predate the index delivery

Yet, the discussion frequently speculates about mechanistic contributors and pathophysiologic pathways without sufficient justification. Adjusting for post-exposure variables (e.g., cardiovascular disease, heart failure, cancer) and then interpreting residual associations as meaningful risks is methodologically inappropriate and risks collider bias and over adjustment. This severely limits causal interpretability.

2. Inappropriate Adjustment Strategy

The fully adjusted mortality models include a very large number of health outcomes that are:

• Likely intermediates on the causal pathway

• Possibly consequences of abnormal placental separation

• Possibly diagnosed after the index delivery

This approach violates core principles of survival analysis and causal inference. Adjusting for downstream consequences of the exposure can:

• Artificially attenuate true associations (as seen with abruption)

• Produce spurious “independent” effects (as claimed for retention)

The authors do not provide a directed acyclic graph (DAG) or any conceptual framework to justify their adjustment set, which is a critical omission given the complexity of the model.

3. Violation of Proportional Hazards Assumptions

The proportional hazards assumption is violated in adjusted models (global p = 0.011), yet Cox models remain the primary analytic approach. Sensitivity analyses excluding early deaths are presented as a remedy, but:

• This is a post hoc solution

• It changes the estimand without clear justification

• It selectively rescues significance for retained placenta while nullifying abruption

More appropriate alternatives (e.g., time-varying coefficients, stratified Cox models, flexible parametric survival models) are not explored. As a result, the core mortality findings are statistically unstable.

4. Small Event Numbers and Sparse Data Bias

Placental abruption and retention comprise ~2% of the cohort combined, yet dozens of covariates and subgroup analyses are conducted. This raises serious concerns about:

• Sparse data bias

• Overfitting

• Inflated type I error due to multiple comparisons

The manuscript does not report:

• Absolute event counts by group

• Events-per-variable ratios

• Any correction for multiple hypothesis testing

This undermines confidence in the reported statistically significant associations, particularly those involving rare outcomes (e.g., thyroid cancer, pulmonary hypertension).

5. Reliance on Administrative Codes Without Validation

The study depends entirely on ICD, CPT, and HCPCS codes to define:

• Placental abruption

• Retained placenta

• Mortality-associated comorbidities

No validation strategy is provided, nor is there discussion of:

• Known coding inaccuracies for obstetric complications

• Differential misclassification between abruption and retention

• Potential bias introduced by care-intensity–dependent coding

The assumption that misclassification would only bias toward the null is unsupported and overly simplistic.

6. Ambiguity in Temporality of “Health Outcomes”

The manuscript does not clearly distinguish between:

• Pre-existing conditions

• Conditions diagnosed during delivery

• Conditions occurring years after delivery

As a result, Table 3 conflates baseline risk factors, acute complications, and long-term sequelae, making interpretation extremely difficult. Without temporal ordering, the associations cannot meaningfully inform mechanism or prognosis.

7. Inconsistent and Confusing Interpretation of Obesity

Obesity is discussed as both a protective factor (in abruption) and a risk factor (in retention), yet:

• BMI data are acknowledged to be incomplete

• Pregnancy weight gain is unavailable

• Obesity is not handled consistently across models

This undermines confidence in obesity-related findings and highlights broader data quality issues within the dataset.

8. Overstated Clinical Implications

The manuscript suggests changes to postpartum monitoring and follow-up based on findings that are:

• Observational

• Statistically fragile

• Heavily model-dependent

PLOS ONE requires conclusions to be strictly supported by the data. The recommendations for targeted surveillance and preventive care are premature and insufficiently justified.

Minor Concerns

• Numerous grammatical errors, missing punctuation, and abrupt transitions detract from readability.

• Citations are sometimes incomplete or incorrectly formatted.

• Tables are referenced extensively but not adequately summarized in the text.

• The conclusion is truncated (“separat…”) suggesting insufficient manuscript proofreading.

Suitability for PLOS ONE

While PLOS ONE does not require novelty, it does require methodological soundness and transparent interpretation. The combination of:

• Over-adjustment

• Violated model assumptions

• Unclear temporality

• Sparse data

• Speculative interpretation

means that the central conclusions are not reliable.

.

Reviewer #1: **Yes:**John M DeSessoJohn M DeSessoJohn M DeSessoJohn M DeSesso

Reviewer #2: No

Reviewer #3: **Yes:**Begum Durkut-KuzuBegum Durkut-KuzuBegum Durkut-KuzuBegum Durkut-Kuzu

Reviewer #4: No

---

## [Author Response · Author response to Decision Letter 1]

7 Feb 2026

Response to the Editor and Reviewers

Manuscript title: A comparison of long-term maternal mortality associated with pathologic placental separation: highlighting possible trends and mechanisms

Dear Editor and Reviewers,

We sincerely thank you for the time and expertise you devoted to reviewing our manuscript. We greatly value the constructive feedback and have undertaken a comprehensive revision that strengthens the clarity, methodologic transparency, and interpretive rigor of the study. Throughout, we have revised the manuscript to ensure that conclusions remain strictly supported by the data, and we have clarified when analyses are descriptive/associational rather than causal.

Summary of major revisions

• Clarified scope and terminology: revised the title and abstract to specify “long-term maternal mortality,” and reduced causal language throughout.

• Improved reproducibility: added explicit ICD-9-CM and ICD-10-CM code definitions for placental abruption and retained placenta, and specified the temporal window used to link diagnostic codes to the index delivery; full code lists are provided in Supporting Information (S1 Appendix).

• Reorganized results presentation: reworked Tables 1–3 to improve readability, to present absolute event counts and rates, and to clearly distinguish baseline characteristics/acute clinical interventions from broader associated diagnoses.

• Strengthened analytic transparency: clarified the Cox model specification and covariate strategy using a DAG-informed approach (primary adjustment set limited to demographic confounders) and added a secondary model that adjusts for selected pre-existing conditions.

• Addressed proportional hazards concerns: evaluated proportional hazards using Schoenfeld residuals (global and variable-specific) and added complementary analyses including time-window–restricted models and piecewise Cox models (Supporting tables).

• Addressed sparse-data and multiplicity concerns: reported events-per-variable for core mortality models (Supporting Information) and applied Bonferroni and Benjamini–Hochberg corrections for primary mortality comparisons; we also explicitly frame secondary associations as exploratory.

• Expanded and tempered interpretation: clarified limitations related to ICD-based definitions, temporality of “health condition” associations, and incomplete BMI data; refined conclusions and clinical implications accordingly.

• Improved writing and formatting: performed thorough proofreading and standardized references/citations to journal requirements.

We thank the reviewers for their thoughtful and constructive comments. We have carefully addressed each point and made substantial revisions to strengthen the manuscript. Below we provide a detailed, point-by-point response to each reviewer. For ease of review, all edits are visible in the revised manuscript with highlighted texts, and we reference key updated tables/appendices where applicable.

Reviewer #1

Comment: The presentation of the copious amount of data within the manuscript’s tables is difficult to understand; please rework tables and explain how data were handled.

Response: We sincerely thank you for the careful and constructive evaluation of our manuscript. We have completed a substantial revision to address your comments and concerns. We agree and have comprehensively restructured Tables 1–3 and expanded the Results narrative to guide readers through the key findings. Specifically: (i) Table 1 is now a clearly labeled baseline characteristics table that additionally reports follow-up time, person-years, and absolute death counts/rates to contextualize subsequent survival analyses; (ii) Table 2 now presents the core mortality hazard ratios with clearly defined adjustment sets and complementary sensitivity analyses to separate short-term vs long-term patterns; (iii) Table 3 has been reframed as a set of adjusted odds ratios from multivariable logistic regression to describe associated diagnoses and we added explicit criteria for how “comparative directionality” is defined. We also expanded the Methods to clarify model specification, covariate selection, and assumption-checking.

Comment: Table 1: It is unclear what the p value refers to; with very different group sizes, statistics seem superfluous. Also, why report year of birth; maternal age would be better.

Response: Thank you—this was an important clarity issue. We revised Table 1 to remove ambiguity and improve clinical interpretability. First, we removed p-values for the cohort size and other descriptive rows where hypothesis testing is not informative; where p-values remain, they now correspond to clearly specified overall group comparisons for that row, and the table header/footnote explains the tests used. Second, we added maternal age at delivery (mean, SD) as a primary demographic descriptor and use it in the primary adjusted mortality model. We also added absolute death counts and mortality rates (per 1,000 deliveries and per 1,000 person-years) to facilitate within-group interpretation rather than relying on significance testing alone.

Comment: SDOH is qualitative; what do the numbers represent, and how does statistical evaluation help?

Response: We thank the reviewer for this comment. In this study, social determinants of health (SDOH) were identified using ICD-10-CM–based social risk indicator codes available in the TriNetX research network. We operationalized SDOH as binary variables (presence vs absence of documented social risk), and the values shown therefore represent the number (%) of patients with at least one recorded SDOH code in each category/group rather than a graded or qualitative measure of social context. We then evaluated whether these coded SDOH indicators, as defined within TriNetX, differed across abnormal placental separation groups and were associated with long-term mortality in adjusted analyses. We recognize that ICD-coded SDOH captures only the portion of social risk that is documented in the medical record and does not reflect the full breadth or nuance of SDOH. To address this, we clarified in the Methods and Table 1 that SDOH in TriNetX is based on ICD-10-CM social risk indicator codes and is analyzed as a binary indicator (presence/absence of any documented social risk code). We also clarified that SDOH is included as a demographic confounder in our primary adjusted mortality model to account for potential differences in documented social risk burden across groups.

Comment: Some entries listed under Health Outcomes are not outcomes (alcohol/tobacco; procedures like transfusion; multiple gestation).

Response: We agree and revised both the labeling and analytic framing. In Table 1, alcohol/tobacco and acute interventions (e.g., transfusion, critical care) are now presented as baseline characteristics and acute clinical factors recorded around the delivery encounter rather than as “health outcomes.” In Table 3, we refer to the broader set of diagnoses as “associated health conditions,” and we explicitly note that, due to database limitations, these associations may reflect conditions present before or after the index delivery. Accordingly, we interpret Table 3 as exploratory associations that may inform hypotheses for future mechanistic and prospective work, rather than as causal or incident outcomes.

Comment: Table 2: Please specify the formula used to determine hazard ratios and how adjustment factors were identified and used.

Response: We expanded the Statistical Analysis section to explicitly describe the Cox proportional hazards regression framework and the interpretation of the hazard ratio as a relative risk of death over follow-up. We also clarified covariate selection using a DAG-informed approach to avoid overadjustment for potential mediators/colliders. Our primary adjusted model controls for demographic confounders (maternal age at delivery, race, and SDOH indicator). A secondary model adds selected pre-existing conditions (diabetes, hypertension, obesity, chronic kidney disease) to evaluate robustness to baseline health status. In addition, Table 2 now includes complementary analyses (censoring at 1 year; excluding deaths ≤42 days; excluding deaths ≤1 year; and piecewise Cox models in Supporting Information) to better characterize temporal patterns.

Comment: Table 3 directionality is opaque; some numerically different values are called “similar,” while closer values are “different.” Discussion does little to clarify inconsistencies.

Response: Thank you for highlighting this. We revised Table 3 in three ways: (i) We changed the analysis from Cox models to multivariable logistic regression and now report adjusted odds ratios (ORs), which is more appropriate for cross-sectional diagnosis associations; (ii) We added a clear definition for the “Comparative Directionality” column: “Similar” indicates both abruption and retention show statistically significant elevated association (p<0.05) in the same direction, whereas “Different” indicates statistical significance for only one condition; (iii) We revised the Results and Discussion to summarize the most clinically salient patterns and to avoid overinterpreting magnitude differences when the aim is to contrast broad association patterns between the two placental conditions.

Reviewer #2

Comment: Some covariates in fully adjusted models could be downstream conditions; please clarify overadjustment potential and implications.

Response: We sincerely thank you for the careful and constructive evaluation of our manuscript. We have completed a substantial revision to address your comments and concerns. We agree. In the original submission, the “fully adjusted” mortality model included a large number of health conditions that could plausibly lie on the causal pathway or reflect post-exposure diagnoses, which could introduce overadjustment and/or collider bias. In the revision, we adopted a DAG-informed strategy and defined a primary adjusted model limited to demographic confounders (maternal age at delivery, race, SDOH indicator). We deliberately avoided adjusting for acute delivery complications and downstream diagnoses in the primary mortality model. We retained a secondary model that adjusts for a small set of pre-existing conditions as a robustness check (not as a causal estimand) and we revised the Discussion to explicitly state that our findings are associational and do not establish causality.

Reviewer #3

Comment: Revise “long-term mortality” to “long-term maternal mortality” in title and abstract.

Response: We sincerely thank you for the careful and constructive evaluation of our manuscript. We have completed a substantial revision to address your comments and concerns. We revised both the title and the abstract to explicitly reference “long-term maternal mortality,” improving immediate clarity for readers.

Comment: Explicitly state ICD codes used to define placental abruption and retained placenta in the Methods.

Response: We added ICD-9-CM and ICD-10-CM code families for placental abruption (ICD-9-CM 641.2x; ICD-10-CM O45.x) and retained placenta (ICD-9-CM 667.0x–667.1x; ICD-10-CM O72.0–O72.2). We also clarified the temporal linkage: placental-separation codes occurring within 6 weeks before to 6 weeks after the delivery date were used to classify the index delivery. Full code lists are provided in Supporting Information (S1 Appendix).

Comment: Expand interpretation of age-related findings in the Discussion.

Response: We expanded the Discussion to contextualize the observed age differences and to emphasize maternal age as a key determinant of baseline health and obstetric risk. We also incorporate maternal age directly into the primary adjusted mortality model and discuss how age-related confounding may contribute to long-term mortality differences.

Reviewer #4

Comment: 1. Conceptual and causal overreach; mechanistic speculation without sufficient justification; inappropriate adjustment for post-exposure variables and interpreting residual associations as causal.

Response: We sincerely thank you for the careful and constructive evaluation of our manuscript. We have completed a substantial revision to address your comments and concerns. We agree with the need to avoid causal overinterpretation. In the revised manuscript, we substantially reduced causal language and reframed mechanistic discussion as hypothesis-generating rather than confirmatory. Critically, we revised the mortality modeling strategy to avoid adjusting for downstream diagnoses/complications in the primary model. We now emphasize that the study is observational and that temporality cannot be fully established for many associated diagnoses in the database. Accordingly, we interpret mortality associations as risk markers and interpret diagnosis associations (Table 3) as exploratory correlates that may inform future mechanistic and prospective validation studies.

Comment: 2. Inappropriate adjustment strategy; lack of DAG or conceptual framework.

Response: We added a DAG-informed covariate selection rationale in the Methods. Our primary adjusted model includes only demographic confounders (maternal age at delivery, race, SDOH indicator). We explicitly avoided adjusting for acute delivery complications (e.g., hemorrhage/DIC) or downstream conditions in the primary mortality model. We include a secondary model with selected pre-existing conditions to assess robustness to baseline health status, and we clarify that this is a sensitivity/robustness analysis rather than a causal adjustment for mediators.

Comment: 3. Proportional hazards assumption violated in adjusted models; suggested alternatives not explored; sensitivity analyses may change estimand.

Response: We agree that proportional hazards assessment and appropriate alternatives are essential. In the revision, we report proportional hazards diagnostics using Schoenfeld residuals and provide both global and variable-specific tests. Importantly, variable-specific testing shows that the exposure (placental separation group) satisfies the PH assumption, while the global violation is driven by adjustment covariates. To further address potential time-varying effects, we added complementary analyses including (i) time-window–restricted models (short-term within 1 year; excluding ≤42-day deaths; excluding ≤1-year deaths) and (ii) piecewise Cox models across prespecified intervals (0–1, 1–2, 2–5, >5 years) presented in Supporting Information. We clarify that time-window analyses evaluate related but distinct estimands and therefore are presented as complementary descriptions of temporal risk patterns rather than “fixes” for the primary model.

Comment: 4. Sparse data bias/overfitting/multiple comparisons; missing event counts, EPV ratios, and correction for multiple testing.

Response: We addressed these concerns in multiple ways. First, we now report absolute death counts by group, follow-up time, and person-years (Table 1 and Results). Second, we quantified events-per-variable (EPV) for core mortality models and demonstrate that EPV is very high for the primary mortality analyses (Supporting Information), mitigating overfitting concerns for those models. Third, for primary mortality outcomes we applied Bonferroni and Benjamini–Hochberg FDR corrections and report corrected inference in Supporting Information. Finally, we explicitly label the broader diagnosis associations (Table 3) as exploratory and interpret them cautiously, emphasizing pattern recognition rather than isolated p-values for rare diagnoses.

Comment: 5. Reliance on administrative codes without validation; misclassification concerns and potential differential coding.

Response: We agree that ICD-based phenotyping is a key limitation. We strengthened the Methods by providing explicit code definitions and the temporal window used to link diagnosis codes to the delivery encounter. We also expanded the Limitations to discuss misclassification risk and cite evidence that prevalence estimates can differ between ICD-based definitions and clinical review. We acknowledge that coding inaccuracies and care-intensity–dependent documentation could introduce bias (including differential misclassification) and that the direction of such bias is not guaranteed. We therefore frame ou

---

## [Decision Letter · Decision Letter 1]

2 Mar 2026

Dear Dr. Demiray,

Thank you for submitting your manuscript to PLOS ONE. After careful consideration, we feel that it has merit but does not fully meet PLOS ONE’s publication criteria as it currently stands. Therefore, we invite you to submit a revised version of the manuscript that addresses the points raised during the review process.

Renaming "health outcomes" to "associated health conditions" was a good move, but the Discussion still occasionally reads as though these associations flow in a particular direction. Given that Table 3 uses cross-sectional logistic regression, there's genuinely no way to know whether these conditions came before or after the delivery. A few sentences in the Discussion need to reflect that more honestly.The limitation around incomplete BMI data is there, but it feels a bit glossed over. Obesity is a serious confounder for both placental complications and long-term cardiovascular death, so a more candid discussion of what that missing data might mean for the mortality findings would strengthen the paper. Even a couple of additional sentences on the likely direction of bias would go a long way.The authors did a solid job pulling back on causal language throughout the revision, but a final careful read is still worth doing. These phrases have a way of surviving in the Discussion and Conclusions where the writing naturally tends toward interpretation. Worth one more pass before the paper goes to production.

We look forward to receiving your revised manuscript.

Kind regards,

Melvin Marzan, BSc, MSc TM, PhD

Academic Editor

PLOS One

Journal Requirements:

Reviewers' comments:

Reviewer's Responses to Questions

**Comments to the Author**

Reviewer #1: All comments have been addressed

Reviewer #2: All comments have been addressed

Reviewer #3: All comments have been addressed

2. Is the manuscript technically sound, and do the data support the conclusions?

Reviewer #1: Yes

Reviewer #2: Yes

Reviewer #3: Yes

3. Has the statistical analysis been performed appropriately and rigorously?

Reviewer #1: Yes

Reviewer #2: Yes

Reviewer #3: Yes

4. Have the authors made all data underlying the findings in their manuscript fully available?

Reviewer #1: Yes

Reviewer #2: Yes

Reviewer #3: Yes

5. Is the manuscript presented in an intelligible fashion and written in standard English?

Reviewer #1: Yes

Reviewer #2: Yes

Reviewer #3: Yes

Reviewer #1: This manuscript is greatly improved. The authors have successfully addressed my comments in both the text and the author responses. I have no further comments.

Reviewer #2: PONE-D-25-61690 R-1

"A comparison of long-term maternal mortality associated with pathologic placental separation: highlighting possible trends and mechanisms"

The revised manuscript has substantially improved in methodological clarity, statistical transparency, and interpretative restraint. The authors have addressed the major concerns, particularly those related to overadjustment, proportional hazards assumptions, sparse data bias, and interpretative overreach.

The revised version demonstrates clear efforts to (1) limit primary adjustment to demographic confounders using a DAG-informed rationale, (2) Separate primary mortality analyses from exploratory diagnostic associations, (3) provide detailed proportional hazards diagnostics, (4) report absolute event counts, person-years, and EPV considerations, (5) Apply appropriate multiplicity correction for primary outcomes, (6) Reframe conclusions in strictly associational terms.

The primary Cox models are now appropriately specified and avoid adjustment for downstream mediators. Sensitivity analyses (time-restricted, exclusion windows, and piecewise Cox modeling) enhance the robustness of the mortality findings. While residual confounding and ICD-based misclassification remain inherent limitations of administrative datasets, these are now clearly acknowledged.

These revisions significantly strengthen the methodological integrity of the study. However, the manuscript could benefit from a brief clarification in the Discussion section, reiterating that the temporality of secondary diagnoses cannot be fully established.

Reviewer #3: I have reviewed the revised manuscript and the authors’ responses. The concerns raised in the previous round have been adequately addressed.

The study is technically sound, the data support the conclusions, and the statistical analyses are appropriate. The manuscript is clearly written and complies with the journal’s data availability requirements.

I have no further comments. The manuscript is suitable for publication in its current form.

.

Reviewer #1: **Yes:**John M DeSessoJohn M DeSessoJohn M DeSessoJohn M DeSesso

Reviewer #2: **Yes:**Dr. Francisco J. Valenzuela-MelgarejoDr. Francisco J. Valenzuela-MelgarejoDr. Francisco J. Valenzuela-MelgarejoDr. Francisco J. Valenzuela-Melgarejo

Reviewer #3: No

---

## [Author Response · Author response to Decision Letter 2]

20 Mar 2026

Dear Dr. Marzan,

Thank you for the opportunity to revise our manuscript. We appreciate the constructive feedback from the editorial team and are pleased that all three reviewers found their previous comments adequately addressed.

We have carefully addressed each of the remaining points:

1. Cross-sectional nature of associations (Table 3)

We have revised the Discussion to more explicitly acknowledge that the temporality of associated health conditions cannot be established given the cross-sectional nature of our logistic regression analyses. We have added clarifying language to ensure readers understand these associations may be bidirectional.

2. Incomplete BMI data and potential confounding

We have expanded our limitations section to provide a more candid discussion of missing BMI data and its implications. We now address the likely direction of bias, given that obesity is a known confounder for both placental complications and long-term cardiovascular mortality.

3. Final review of causal language

We conducted a thorough final review of the Discussion and Conclusions sections to remove any remaining causal language, ensuring all findings are presented in strictly associational terms.

All changes are highlighted in the attached "Revised Manuscript with Track Changes" document. We have also verified our reference list is complete and current.

We thank the reviewers for their thoughtful evaluation and confirmation that the manuscript is suitable for publication. We believe these final revisions strengthen the manuscript's interpretative rigor.

Sincerely,

Atalay Demiray, MD, MSc

(on behalf of all authors)

---

## [Editor Report · Decision Letter 2]

7 Apr 2026

A comparison of long-term maternal mortality associated with pathologic placental separation: highlighting possible trends and mechanisms

PONE-D-25-61690R2

Dear Dr. Demiray,

We’re pleased to inform you that your manuscript has been judged scientifically suitable for publication and will be formally accepted for publication once it meets all outstanding technical requirements.

Kind regards,

Melvin Marzan, BSc, MSc TM, PhD

Academic Editor

PLOS One
---

## [Editor Report · Acceptance letter]

PONE-D-25-61690R2

PLOS One

Dear Dr. Demiray,

I'm pleased to inform you that your manuscript has been deemed suitable for publication in PLOS One. Congratulations! Your manuscript is now being handed over to our production team.

Kind regards,

on behalf of

Dr. Melvin Marzan

Academic Editor

PLOS One